# Unsupervised Data Augmentation for Consistency Training

## Abstract

Semi-supervised learning lately has shown much promise in improving deep learning models when labeled data is scarce. Common among recent approaches is the use of consistency training on a large amount of unlabeled data to constrain model predictions to be invariant to input noise. In this work, we present a new perspective on how to effectively noise unlabeled examples and argue that the quality of noising, specifically those produced by advanced data augmentation methods, plays a crucial role in semi-supervised learning. By substituting simple noising operations with advanced data augmentation methods, our method brings substantial improvements across six language and three vision tasks under the same consistency training framework. On the IMDb text classification dataset, with only 20 labeled examples, our method achieves an error rate of 4.20, outperforming the state-of-the-art model trained on 25,000 labeled examples. On a standard semi-supervised learning benchmark, CIFAR-10, our method outperforms all previous approaches and achieves an error rate of 2.7% with only 4,000 examples, nearly matching the performance of models trained on 50,000 labeled examples. Our method also combines well with transfer learning, e.g., when finetuning from BERT, and yields improvements in high-data regime, such as ImageNet, whether when there is only 10% labeled data or when a full labeled set with 1.3M extra unlabeled examples is used. [1]

## 1 Introduction

A fundamental weakness of deep learning is that it typically requires a lot of labeled data to work well. Semi-supervised learning (SSL) (Chapelle et al., 2009) is one of the most promising methods of leveraging unlabeled data to address this weakness. The recent works in SSL are diverse but those that are based on consistency training (Bachman et al., 2014; Rasmus et al., 2015; Laine & Aila, 2016; Tarvainen & Valpola, 2017) have shown to work well on many benchmarks.

In a nutshell, consistency training methods simply regularize model predictions to be invariant to small noise applied to either input examples (Miyato et al., 2018; Sajjadi et al., 2016; Clark et al., 2018) or hidden states (Bachman et al., 2014; Laine & Aila, 2016). This framework makes sense intuitively because a good model should be robust to any small change in an input example or hidden states. Under this framework, different methods in this category differ mostly in how and where the noise injection is applied. Typical noise injection methods are additive Gaussian noise, dropout noise or adversarial noise.

In this work, we investigate the role of noise injection in consistency training and observe that advanced data augmentation methods, specifically those work best in supervised learning (Simard et al., 1998; Krizhevsky et al., 2012; Cubuk et al., 2018; Yu et al., 2018), also perform well in semi-supervised learning. There is indeed a strong correlation between the performance of data augmentation operations in supervised learning and their performance in consistency training. We, hence, propose to substitute the traditional noise injection methods with high quality data augmentation methods in order to improve consistency training. To emphasize the use of better data augmentation in consistency training, we name our method Unsupervised Data Augmentation or UDA.

---

[1]Code is available at an anonymous link.

We evaluate UDA on a wide variety of language and vision tasks. On six text classification tasks, our method achieves significant improvements over state-of-the-art models. Notably, on IMDb, UDA with 20 labeled examples outperforms the state-of-the-art model trained on 1250x more labeled data. We also evaluate UDA on standard semi-supervised learning benchmarks for vision such as CIFAR-10 and SVHN. UDA outperforms all existing semi-supervised learning methods by significant margins. On CIFAR-10 with 4,000 labeled examples, UDA achieves an error rate of 5.29, nearly matching the performance of the fully supervised model that uses 50,000 labeled examples. Furthermore, with a better architecture, PyramidNet+ShakeDrop, UDA achieves a new state-of-the-art error rate of 2.7. On SVHN, UDA achieves an error rate of 2.55 with only 1,000 labeled examples. Finally, we also find UDA to be beneficial when there is a large amount of supervised data. For instance, on ImageNet, UDA leads to improvements of top-1 accuracy from $58.84$ to $68.78$ with $10\%$ of the labeled set and from $78.43$ to $79.05$ when we use the full labeled set and an external dataset with $1.3M$ unlabeled examples.

Our key contributions and findings can be summarized as follows:

- First, we show that state-of-the-art data augmentations found in supervised learning can also serve as a superior source of perturbation under the consistency enforcing semi-supervised framework. *See results in Table 1 and Table 2.*

- Second, we show that UDA can match and even outperform purely supervised learning that uses orders of magnitude more labeled data.

  *State-of-the-art results for both vision and language tasks are reported in Table 3 and 4. The effectiveness of UDA across different training data sizes are highlighted in Figure 4 and 5.*

- Finally, we show that UDA combines well with transfer learning, e.g., when fine-tuning from BERT (*see Table 4*), and is effective at high-data regime, e.g. on ImageNet (*see Table 5*).

## 2 UNSUPERVISED DATA AUGMENTATION (UDA)

In this section, we first formulate our task and then present the key method and insights behind UDA. Throughout this paper, we focus on classification problems and will use $x$ to denote the input and $y^*$ to denote its ground-truth prediction target. We are interested in learning a model $p_\theta(y \mid x)$ to predict $y^*$ based on the input $x$, where $\theta$ denotes the model parameters. Finally, we will use $L$ and $U$ to denote the sets of labeled and unlabeled examples respectively.

### 2.1 BACKGROUND: SUPERVISED DATA AUGMENTATION

Data augmentation aims at creating novel and realistic-looking training data by applying a transformation to an example, without changing its label. Formally, let $q(\hat{x} \mid x)$ be the augmentation transformation from which one can draw augmented examples $\hat{x}$ based on an original example $x$. For an augmentation transformation to be valid, it is required that any example $\hat{x} \sim q(\hat{x} \mid x)$ drawn from the distribution shares the same ground-truth label as $x$. Given a valid augmentation transformation, we can simply minimize the negative log-likelihood on augmented examples.

Supervised data augmentation can be equivalently seen as constructing an augmented labeled set from the original supervised set and then training the model on the augmented set. Therefore, the augmented set needs to provide additional inductive biases to be more effective. How to design the augmentation transformation has, thus, become critical.

In recent years, there have been significant advancements on the design of data augmentations for NLP (Yu et al., 2018), vision (Krizhevsky et al., 2012; Cubuk et al., 2018) and speech (Hannun et al., 2014; Park et al., 2019) in supervised settings. Despite the promising results, data augmentation is mostly regarded as the "cherry on the cake" which provides a steady but limited performance boost because these augmentations has so far only been applied to a set of labeled examples which is usually of a small size. Motivated by this limitation, via the consistency training framework, we extend the advancement in supervised data augmentation to semi-supervised learning where abundant unlabeled data is available.

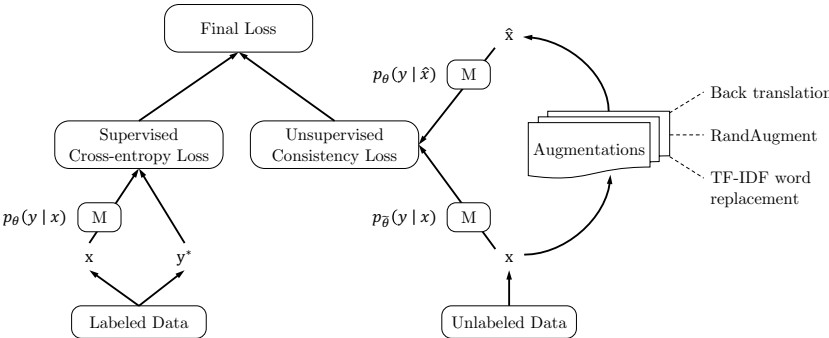

Figure 1: Training objective for UDA, where M is a model that predicts a distribution of $y$ given $x$.

## 2.2 UNSUPERVISED DATA AUGMENTATION

As discussed in the introduction, a recent line of work in semi-supervised learning has been utilizing unlabeled examples to enforce smoothness of the model. The general form of these works can be summarized as follows:

- Given an input $x$, compute the output distribution $p_\theta(y \mid x)$ given $x$ and a noised version $p_\theta(y \mid x, \epsilon)$ by injecting a small noise $\epsilon$. The noise can be applied to $x$ or hidden states.
- Minimize a divergence metric between the two distributions $\mathcal{D}\left(p_\theta(y \mid x) \parallel p_\theta(y \mid x, \epsilon)\right)$.

This procedure enforces the model to be insensitive to the noise $\epsilon$ and hence smoother with respect to changes in the input (or hidden) space. From another perspective, minimizing the consistency loss gradually propagates label information from labeled examples to unlabeled ones.

In this work, we are interested in a particular setting where the noise is injected to the input $x$, i.e., $\hat{x} = q(x, \epsilon)$, as considered by prior works (Sajjadi et al., 2016; Laine & Aila, 2016; Miyato et al., 2018). But different from existing work, we focus on the unattended question of how the form or "quality" of the noising operation $q$ can influence the performance of this consistency training framework. Specifically, to enforce consistency, prior methods generally employ simple noise injection methods such as adding Gaussian noise, simple input augmentations to noise unlabeled examples. In contrast, we hypothesize that stronger data augmentations in supervised learning can also lead to superior performance when used to noise unlabeled examples in the semi-supervised consistency training framework, since it has been shown that more advanced data augmentations that are more diverse and natural can lead to significant performance gain in the supervised setting.

Following this idea, we propose to use a rich set of state-of-the-art data augmentations verified in various supervised settings to inject noise and optimize the same consistency training objective on unlabeled examples. When jointly trained with labeled examples, we utilize a weighting factor $\lambda$ to balance the supervised cross entropy and the unsupervised consistency training loss, which is illustrated in Figure 1. Formally, the full objective can be written as follows:

$$\min_\theta \ \mathcal{J}(\theta) = \mathbb{E}_{x, y^* \in L}\left[-\log p_\theta(y^* \mid x)\right] + \lambda \mathbb{E}_{x \in U} \mathbb{E}_{\hat{x} \sim q(\hat{x}|x)}\left[\mathcal{D}_{\mathrm{KL}}\left(p_{\tilde{\theta}}(y \mid x) \parallel p_\theta(y \mid \hat{x}))\right)\right].$$

where $q(\hat{x} \mid x)$ is a data augmentation transformation and $\tilde{\theta}$ is a *fixed* copy of the current parameters $\theta$ indicating that the gradient is not propagated through $\tilde{\theta}$, as suggested by Miyato et al. (2018). We also follow VAT (Miyato et al., 2018) to use the KL divergence. We set $\lambda$ to 1 for most of our experiments and use different batch sizes for the supervised data and the unsupervised data. In the vision domain, simple augmentations including cropping and flipping are applied to labeled examples. To minimize the discrepancy between supervised training and prediction on unlabeled examples, we apply the same simple augmentations to unlabeled examples for computing $p_{\tilde{\theta}}(y \mid x)$.

**Discussion.** Before detailing the augmentation operations used in this work, we first provide some intuitions on how more advanced data augmentations can provide extra advantages over simple ones used in earlier works from three aspects:

- **Valid noise**: Advanced data augmentation methods that achieve great performance in supervised learning usually generate realistic augmented examples that share the same ground-truth labels

with the original example. Thus, it is safe to encourage the consistency between predictions on the original unlabeled example and the augmented unlabeled examples.

- **Diverse noise**: Advanced data augmentation can generate a diverse set of examples since it can make large modifications to the input example without changing its label, while simple Gaussian noise only make local changes. Encouraging consistency on a diverse set of augmented examples can significantly improve the sample efficiency.

- **Targeted inductive biases**: Different tasks require different inductive biases. Data augmentation operations that work well in supervised training essentially provides the missing or most wanted inductive biases in an original labeled set.

## 2.3 AUGMENTATION STRATEGIES FOR DIFFERENT TASKS

We now detail the augmentation methods, tailored for different tasks, that we use in this work.

**RandAugment for Image Classification.** We make use of a data augmentation method called RandAugment, which is inspired by AutoAugment (Cubuk et al., 2018). AutoAugment uses a search method to combine all image processing transformations in the Python Image Library (PIL) to find a good augmentation strategy. In RandAugment, we do not use search, but instead uniformly sample from the same set of augmentation transformations in PIL. In other words, RandAugment is simpler and requires no labeled data as there is no need to search for optimal policies.

**Back-translation for Text Classification.** When used as an augmentation method, back-translation (Sennrich et al., 2015; Edunov et al., 2018) refers to the procedure of translating an existing example $x$ in language $A$ into another language $B$ and then translating it back into $A$ to obtain an augmented example $\hat{x}$. As observed by Yu et al. (2018), back-translation can generate diverse paraphrases while preserving the semantics of the original sentences, leading to significant performance improvements in question answering. In our case, we use back-translation to paraphrase the training data of our text classification tasks.[2]

We find that the diversity of the paraphrases is more important than the quality or the validity. Hence, we employ random sampling with a tunable temperature instead of beam search for the generation. As shown in Figure 2, the paraphrases generated by back-translation sentence are diverse and have similar semantic meanings. More specifically, we use WMT'14 English-French translation models (in both directions) to perform back-translation on each sentence. To facilitate future research, we have open-sourced our back-translation system together with the translation checkpoints.

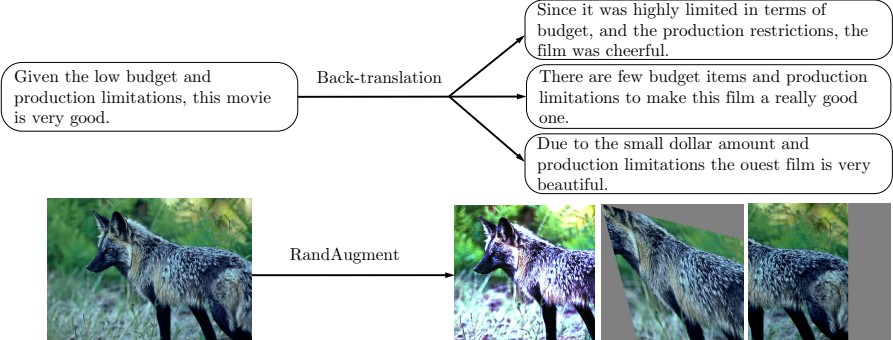

Figure 2: Augmented examples using back-translation and RandAugment.

**Word replacing with TF-IDF for Text Classification.** While back-translation is good at maintaining the global semantics of a sentence, there is little control over which words will be retained. This requirement is important for topic classification tasks, such as DBPedia, in which some keywords are more informative than other words in determining the topic. We, therefore, propose an augmen-

---

[2]We also note that while translation uses a labeled dataset, the translation task itself is quite distinctive from a text classification task and does not make use of any text classification label. In addition, back-translation is a general data augmentation method that can be applied to many tasks with the same model checkpoints.

tation method that replaces uninformative words with low TF-IDF scores while keeping those with high TF-IDF values. We refer readers to Appendix C for a detailed description.

## 2.4 TRAINING SIGNAL ANNEALING FOR LOW-DATA REGIME

In semi-supervised learning, we often encounter a situation where there is a huge gap between the amount of unlabeled data and that of labeled data. Hence, the model often quickly overfits the limited amount of labeled data while still underfitting the unlabeled data. To tackle this difficulty, we introduce a new training technique, called Training Signal Annealing (TSA), which gradually releases the "training signals" of the labeled examples as training progresses. Intuitively, we only utilize a labeled example if the model's confidence on that example is lower than a predefined threshold which increases according to a schedule. Specifically, at training step $t$, if the model's predicted probability for the correct category $p_\theta(y^* \mid x)$ is higher than a threshold $\eta_t$, we remove that example from the loss function. Suppose $K$ is the number of categories, by gradually increase $\eta_t$ from $\frac{1}{K}$ to 1, the threshold $\eta_t$ serves as a ceiling to prevent over-training on easy labeled examples.

We consider three increasing schedules of $\eta_t$ with different application scenarios. Let $T$ be the total number of training steps, the three schedules are shown in Figure 3. Intuitively, when the model is prone to overfit, e.g., when the problem is relatively easy or the number of labeled examples is very limited, the exp-schedule is most suitable as the supervised signal is mostly released at the end of training. In contrast, when the model is less likely to overfit (e.g., when we have abundant labeled examples or when the model employs effective regularization), the log-schedule can serve well.

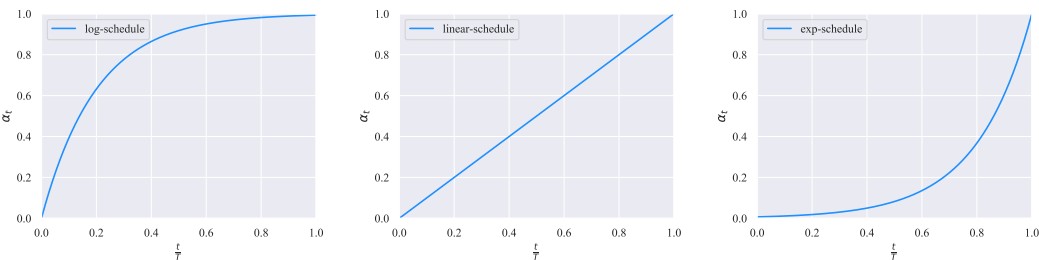

Figure 3: Three schedules of TSA. We set $\eta_t = \alpha_t * (1 - \frac{1}{K}) + \frac{1}{K}$. $\alpha_t$ is set to $1 - \exp(-\frac{t}{T} * 5)$, $\frac{t}{T}$ and $\exp((\frac{t}{T} - 1) * 5)$ for the log, linear and exp schedules.

## 3 EXPERIMENTS

In this section, we evaluate UDA on a variety of language and vision tasks. For language, we rely on six text classification benchmark datasets, including IMDb, Yelp-2, Yelp-5, Amazon-2 and Amazon-5 sentiment classification and DBPedia topic classification (Maas et al., 2011; Zhang et al., 2015). For vision, we employ two smaller datasets CIFAR-10 (Krizhevsky & Hinton, 2009), SVHN (Netzer et al., 2011), which are often used to compare semi-supervised algorithms, as well as ImageNet (Deng et al., 2009) of a larger scale to test the scalability of UDA. For details of the labeled and unlabeled data and experiment details, we refer readers to Appendix E.

### 3.1 CORRELATION BETWEEN SUPERVISED AND SEMI-SUPERVISED PERFORMANCES

As the first step, we try to verify the fundamental idea of UDA, i.e., there is a positive correlation of data augmentation's effectiveness in supervised learning and semi-supervised learning. Based on Yelp-5 (a language task) and CIFAR-10 (a vision task), we compare the performance of different data augmentation methods in either fully supervised or semi-supervised settings. For Yelp-5, apart from back-translation, we include a simpler method Switchout (Wang et al., 2018) which replaces a token with a random token uniformly sampled from the vocabulary. For CIFAR-10, we compare RandAugment with two simpler methods: (1) cropping & flipping augmentation and (2) Cutout.

Based on this setting, Table 1 and Table 2 exhibit a strong correlation of an augmentation's effectiveness between supervised and semi-supervised settings. This validates our idea of stronger

data augmentations found in supervised learning can always lead to more gains when applied to the semi-supervised learning settings.

| Augmentation (# Sup examples) | Sup (50k) | Semi-Sup (4k) |
|---|---|---|
| Crop & flip | 5.36 | 16.17 |
| Cutout | 4.42 | 6.42 |
| RandAugment | **4.23** | **5.29** |

Table 1: Error rates on CIFAR-10.

| Augmentation (# Sup examples) | Sup (650k) | Semi-sup (2.5k) |
|---|---|---|
| ✗ | 38.36 | 50.80 |
| Switchout | 37.24 | 43.38 |
| Back-translation | **36.71** | **41.35** |

Table 2: Error rate on Yelp-5.

## 3.2 ALGORITHM COMPARISON ON VISION SEMI-SUPERVISED LEARNING BENCHMARKS

With the correlation established above, the next question we ask is how well UDA performs compared to existing semi-supervised learning algorithms. To answer the question, we focus on the most commonly used semi-supervised learning benchmarks CIFAR-10 and SVHN.

**Vary the size of labeled data.** Firstly, we follow the settings in (Oliver et al., 2018) and employ Wide-ResNet-28-2 (Zagoruyko & Komodakis, 2016; He et al., 2016) as the backbone model and evaluate UDA with varied supervised data sizes. Specifically, we compare UDA with two highly competitive baselines: (1) Virtual adversarial training (VAT) (Miyato et al., 2018), an algorithm that generates adversarial Gaussian noise on input, and (2) MixMatch (Berthelot et al., 2019), a parallel work that combines previous advancements in semi-supervised learning. The comparison is shown in Figure 4 with two key observations.[3]

- First, UDA consistently outperforms the two baselines with a clear margin given different sizes of labeled data.
- Moreover, the performance difference between UDA and VAT shows the superiority of data augmentation based noise. The difference of UDA and VAT is essentially the noise process. While the noise produced by VAT often contain high-frequency artifacts that do not exist in real images, data augmentation mostly generates diverse and realistic images.

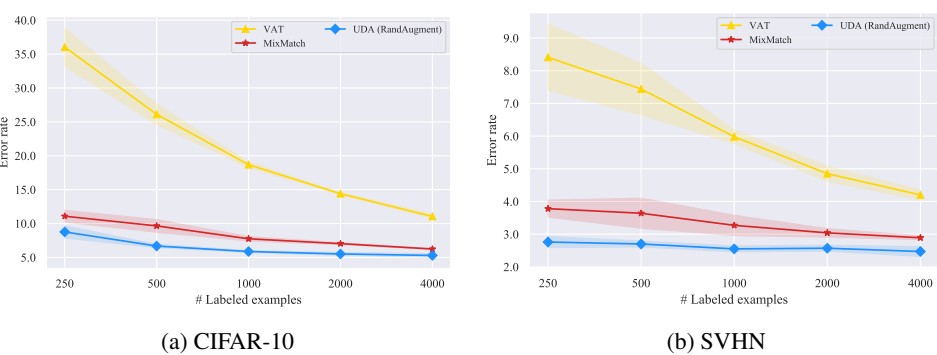

(a) CIFAR-10                    (b) SVHN

Figure 4: Comparison with two semi-supervised learning methods on CIFAR-10 and SVHN with varied number of labeled examples.

**Comparisons with published results** Next, we directly compare UDA with previously published results under different model architectures. Following previous work, 4k and 1k labeled examples are used for CIFAR-10 and SVHN respectively. As shown in Table 3, given the same architecture, UDA outperforms all published results by significant margins. This shows the huge potential of state-of-the-art data augmentations under the consistency training framework in the vision domain.

---

[3]Please refer to Appendix E.2 for detailed hyper-parameters. We only use a different hyper-parameter for the case of 250 examples on CIFAR-10. The hyperparameters for other data sizes are the same. For the case with 250 examples on CIFAR-10, applying hyperparameters used in other data sizes leads to an error rate of $16.84 \pm 4.19$, which might be resulted from a stability issue.

| Method | Model | # Param | CIFAR-10 (4k) | SVHN (1k) |
|---|---|---|---|---|
| Π-Model (Laine & Aila, 2016) | Conv-Large | 3.1M | $12.36 \pm 0.31$ | $4.82 \pm 0.17$ |
| Mean Teacher (Tarvainen & Valpola, 2017) | Conv-Large | 3.1M | $12.31 \pm 0.28$ | $3.95 \pm 0.19$ |
| VAT + EntMin (Miyato et al., 2018) | Conv-Large | 3.1M | $10.55 \pm 0.05$ | $3.86 \pm 0.11$ |
| SNTG (Luo et al., 2018) | Conv-Large | 3.1M | $10.93 \pm 0.14$ | $3.86 \pm 0.27$ |
| VAdD (Park et al., 2018) | Conv-Large | 3.1M | $11.32 \pm 0.11$ | $4.16 \pm 0.08$ |
| Fast-SWA (Athiwaratkun et al., 2018) | Conv-Large | 3.1M | 9.05 | - |
| ICT (Verma et al., 2019) | Conv-Large | 3.1M | $7.29 \pm 0.02$ | $3.89 \pm 0.04$ |
| Pseudo-Label (Lee, 2013) | WRN-28-2 | 1.5M | $16.21 \pm 0.11$ | $7.62 \pm 0.29$ |
| LGA + VAT (Jackson & Schulman, 2019) | WRN-28-2 | 1.5M | $12.06 \pm 0.19$ | $6.58 \pm 0.36$ |
| mixmixup (Hataya & Nakayama, 2019) | WRN-28-2 | 1.5M | 10 | - |
| ICT (Verma et al., 2019) | WRN-28-2 | 1.5M | $7.66 \pm 0.17$ | $3.53 \pm 0.07$ |
| MixMatch (Berthelot et al., 2019) | WRN-28-2 | 1.5M | $6.24 \pm 0.06$ | $2.89 \pm 0.06$ |
| Mean Teacher (Tarvainen & Valpola, 2017) | Shake-Shake | 26M | $6.28 \pm 0.15$ | - |
| Fast-SWA (Athiwaratkun et al., 2018) | Shake-Shake | 26M | 5.0 | - |
| MixMatch (Berthelot et al., 2019) | WRN | 26M | $4.95 \pm 0.08$ | - |
| UDA (RandAugment) | WRN-28-2 | 1.5M | $5.29 \pm 0.25$ | $\mathbf{2.55 \pm 0.09}$ |
| UDA (RandAugment) | Shake-Shake | 26M | 3.7 | - |
| UDA (RandAugment) | PyramidNet | 26M | **2.7** | - |

Table 3: Comparison between methods using different models where PyramidNet is used with ShakeDrop regularization. Fully supervised Wide-ResNet-28-2 and PyramidNet+ShakeDrop have an error rate of 5.4 and 2.7 when trained on 50,000 examples without RandAugment. On CIFAR-10, with only 4,000 labeled examples, UDA matches the performance of the two fully supervised models. On SVHN, UDA also matches the performance of our fully supervised model trained on 73,257 examples without RandAugment, which has an error rate of 2.84.

## 3.3 EVALUATION ON TEXT CLASSIFICATION DATASETS

Next, we further evaluate UDA in the language domain. Moreover, in order to test whether UDA can be combined with the success of unsupervised representation learning, such as BERT (Devlin et al., 2018), we further consider four initialization schemes: (a) random Transformer; (b) $BERT_{BASE}$; (c) $BERT_{LARGE}$; (d) $BERT_{FINETUNE}$: $BERT_{LARGE}$ fine-tuned on in-domain unlabeled data[4]. Under each of these four initialization schemes, we compare the performances with and without UDA.

The results are presented in Table 4 where we would like to emphasize three observations:

- First, even with very few labeled examples, UDA can offer decent or even competitive performances compared to the SOTA model trained with full supervised data. Particularly, on binary sentiment analysis tasks, with only 20 supervised examples, UDA outperforms the previous SOTA trained with full supervised data on IMDb and is competitive on Yelp-2 and Amazon-2.

- Second, UDA is complementary to transfer learning / representation learning. As we can see, when initialized with BERT and further finetuned on in-domain data, UDA can still significantly reduce the error rate from $6.50$ to $4.20$ on IMDb.

- Finally, we also note that for five-category sentiment classification tasks, there still exists a clear gap between UDA with 500 labeled examples per class and BERT trained on the entire supervised set. Intuitively, five-category sentiment classifications are much more difficult than their binary counterparts. This suggests a room for further improvement in the future.

**Results with different labeled set sizes.** We also show in Figure 5 that UDA leads to consistent improvements across all labeled data sizes on IMDb and Yelp-2.

---

[4]One exception is that we do not pursue $BERT_{FINETUNE}$ on DBPedia as fine-tuning BERT on DBPedia does not yield further performance gain. This is probably due to the fact that DBPedia is based on Wikipedia while BERT is already trained on the whole Wikipedia corpus.

| **Fully supervised baseline** | | | | | | |
|---|---|---|---|---|---|---|
| **Datasets** (# Sup examples) | IMDb (25k) | Yelp-2 (560k) | Yelp-5 (650k) | Amazon-2 (3.6m) | Amazon-5 (3m) | DBpedia (560k) |
| Pre-BERT SOTA | *4.32* | 2.16 | 29.98 | 3.32 | 34.81 | 0.70 |
| BERT$_{\text{LARGE}}$ | 4.51 | *1.89* | *29.32* | *2.63* | *34.17* | *0.64* |
| **Semi-supervised setting** | | | | | | |
| **Initialization** | **UDA** | IMDb (20) | Yelp-2 (20) | Yelp-5 (2.5k) | Amazon-2 (20) | Amazon-5 (2.5k) | DBpedia (140) |
| Random | ✗ | 43.27 | 40.25 | 50.80 | 45.39 | 55.70 | 41.14 |
| | ✓ | 25.23 | 8.33 | 41.35 | 16.16 | 44.19 | 7.24 |
| BERT$_{\text{BASE}}$ | ✗ | 18.40 | 13.60 | 41.00 | 26.75 | 44.09 | 2.58 |
| | ✓ | 5.45 | 2.61 | 33.80 | 3.96 | 38.40 | 1.33 |
| BERT$_{\text{LARGE}}$ | ✗ | 11.72 | 10.55 | 38.90 | 15.54 | 42.30 | 1.68 |
| | ✓ | 4.78 | 2.50 | 33.54 | 3.93 | 37.80 | 1.09 |
| BERT$_{\text{FINETUNE}}$ | ✗ | 6.50 | 2.94 | 32.39 | 12.17 | 37.32 | - |
| | ✓ | **4.20** | **2.05** | **32.08** | **3.50** | **37.12** | - |

Table 4: Error rates on text classification datasets. In the fully supervised settings, the pre-BERT SO-TAs include ULMFiT (Howard & Ruder, 2018) for Yelp-2 and Yelp-5, DPCNN (Johnson & Zhang, 2017) for Amazon-2 and Amazon-5, Mixed VAT (Sachan et al., 2018) for IMDb and DBPedia. All of our experiments use a sequence length of 512.

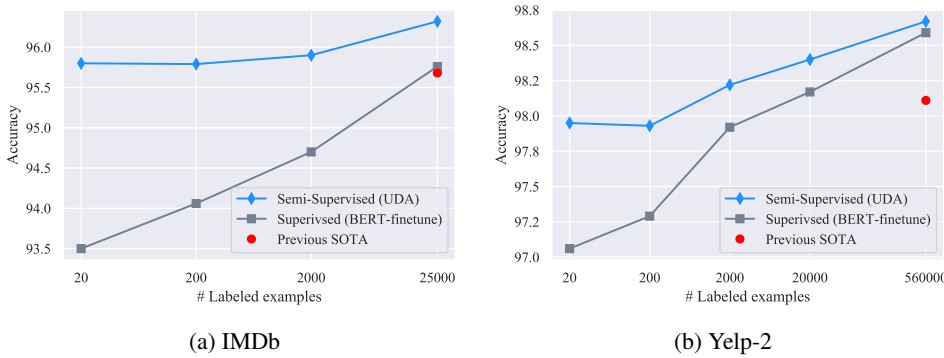

(a) IMDb

(b) Yelp-2

Figure 5: Accuracy on IMDb and Yelp-2 with different number of labeled examples. In the large-data regime, with the full training set of IMDb, UDA also provides robust gains.

## 3.4 SCALABILITY TEST ON THE IMAGENET DATASET

Then, to evaluate whether UDA can scale to problems with a large scale and a higher difficulty, we now turn to the ImageNet dataset with ResNet-50 being the underlying architecture. Specifically, we consider two experiment settings with different natures:

- We use 10% of the supervised data of ImageNet while using all other data as unlabeled data. As a result, the unlabeled exmaples are entirely in-domain.
- In the second setting, we keep all images in ImageNet as supervised data. Then, we use the domain-relevance data filtering method (See Appendix B for details) to filter out 1.3M images from an anonymous dataset. Hence, the unlabeled set is not necessarily in-domain.

The results are summarized in Table 5. In both 10% and the full data settings, UDA consistently brings significant gains compared to the supervised baseline. This shows UDA is not only able to scale but also able to utilize out-of-domain unlabeled examples to improve model performance. In parallel to our work, S4L (Zhai et al., 2019b) and CPC (Hénaff et al., 2019) also show significant improvements on ImageNet.

| Methods | SSL | 10% | 100% |
|---------|-----|-----|------|
| ResNet-50 | ✗ | 55.09 / 77.26 | 77.28 / 93.73 |
| w. RandAugment | | 58.84 / 80.56 | 78.43 / 94.37 |
| UDA (RandAugment) | ✓ | **68.78 / 88.80** | **79.05 / 94.49** |

Table 5: Top-1 / top-5 accuracy on ImageNet with 10% and 100% of the labeled set. We use image size 224 and 331 for the 10% and 100% experiments respectively.

### 3.5 ABLATION STUDIES FOR TSA

Lastly, we study the effect of TSA on two tasks with different amounts of unlabeled data: (a) Yelp-5 where we have only 2.5k labeled examples and 6m unlabeled examples. (b) CIFAR-10 where we have 4k labeled examples and 50k unlabeled examples. For Yelp-5, we use a randomly initialized transformer in this study to rule out factors of having a pre-trained representation.

As shown in Table 6, on Yelp-5, where there is a lot more unlabeled data than labeled data, TSA reduces the error rate from $50.81$ to $41.35$ when compared to the baseline without TSA. More specifically, the best performance is achieved when we choose to postpone releasing the supervised training signal to the end of the training, i.e, exp-schedule leads to the best performance. On the other hand, linear-schedule is the sweet spot on CIFAR-10 in terms of the speed of releasing supervised training signals, where the amount of unlabeled data is comparable to that of supervised data.

| TSA schedule | Yelp-5 | CIFAR-10 |
|--------------|--------|----------|
| ✗ | 50.81 | 5.67 |
| log-schedule | 49.06 | 5.67 |
| linear-schedule | 45.41 | **5.29** |
| exp-schedule | **41.35** | 7.81 |

Table 6: Ablation study for Training Signal Annealing (TSA) on Yelp-5 and CIFAR-10. The shown numbers are error rates.

## 4 RELATED WORK

Existing works in consistency training does make use of data augmentation (Laine & Aila, 2016; Sajjadi et al., 2016); however, they only apply weak augmentation methods such as random translations and cropping. In parallel to our work, ICT (Verma et al., 2019) and MixMatch (Berthelot et al., 2019) also show improvements for semi-supervised learning. These methods employ mixup (Zhang et al., 2017) on top of simple augmentations such as flipping and cropping; instead, UDA emphasizes on the use of state-of-the-art data augmentations, leading to significantly better results on CIFAR-10 and SVHN. In addition, UDA is also applicable to language domain and can also scale well to more challenging vision datasets, such as ImageNet.

Other works in the consistency training family mostly differ in how the noise is defined: Pseudo-ensemble (Bachman et al., 2014) directly applies Gaussian noise and Dropout noise; VAT (Miyato et al., 2018; 2016) defines the noise by approximating the direction of change in the input space that the model is most sensitive to; Cross-view training (Clark et al., 2018) masks out part of the input data. Apart from enforcing consistency on the input examples and the hidden representations, another line of research enforces consistency on the model parameter space. Works in this category include Mean Teacher (Tarvainen & Valpola, 2017), fast-Stochastic Weight Averaging (Athiwaratkun et al., 2018) and Smooth Neighbors on Teacher Graphs (Luo et al., 2018). For a complete version of related work, see Appendix D.

## 5 CONCLUSION

In this paper, we show that data augmentation and semi-supervised learning are well connected: better data augmentation can lead to significantly better semi-supervised learning. Our method,

UDA, employs state-of-the-art data augmentation found in supervised learning to generate diverse and realistic noise and enforces the model to be consistent with respect to these noise. For text, UDA combines well with representation learning, e.g., BERT, and is very effective in low-data regime where state-of-the-art performance is achieved on IMDb with only 20 examples. For vision, UDA outperforms prior works by a clear margin and nearly matches the performance of the fully supervised models trained on the full labeled sets which are one order of magnitude larger. Lastly, UDA can effectively leverage out-of-domain unlabeled data and achieve improved performances on ImageNet where we have a large amount of supervised data. We hope that UDA will encourage future research to transfer advanced supervised augmentation to semi-supervised setting for different tasks.

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

## A   MORE EXPERIMENTS

### A.1   ABLATIONS STUDIES ON RANDAUGMENT

We hypothesize that the success of RandAugment should be credited to the diversity of the augmentation transformations, since RandAugment works very well for multiple different datasets while does not require a search algorithm to find out the most effective policies. To verify this hypothesis, we test UDA's performance when we restrict the number of possible transformations used in RandAugment. As shown in Figure 6, the performance gradually improves as we use more augmentation transformations.

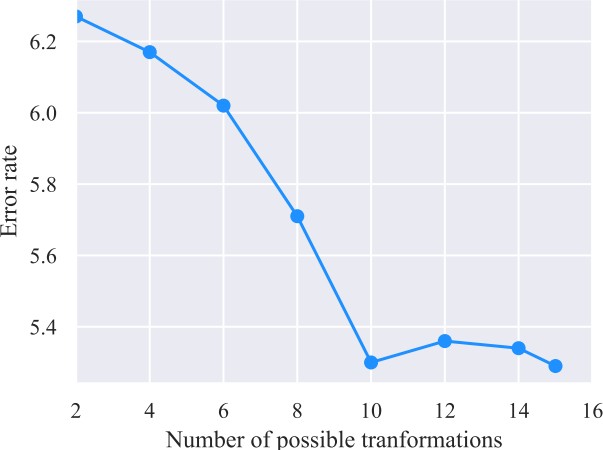

Figure 6: Error rate of UDA on CIFAR-10 with different numbers of possible transformations in RandAugment. UDA achieves lower error rate when we increase the number of possible transformations, which demonstrates the importance of a rich set of augmentation transformations.

### A.2   RESULTS ON CIFAR-10 AND SVHN WITH VARIED LABEL SET SIZES

**CIFAR-10**   In Table 7, we show results for compared methods of Figure 4a and results of Pseudo-Label (Lee, 2013), Π-Model (Laine & Aila, 2016), Mean Teacher (Tarvainen & Valpola, 2017). Fully supervised learning using 50,000 examples achieves an error rate of 5.36 and 4.23 with or without RandAugment. The performance of the baseline models are reported by MixMatch (Berthelot et al., 2019).

To make sure that the performance reported by MixMatch and our results are comparable, we reimplement MixMatch in our codebase and find that the results in the original paper is comparable but slightly higher than our reimplementation, which results in a more competitive comparison for UDA. For example, our reimplementation of MixMatch achieves an error rate of $7.00 \pm 0.59$ and $7.39 \pm 0.11$ with 4,000 and 2,000 examples. MixMatch uses a different model implementation and employs exponential moving average (EMA) on the model parameters, while we do not use EMA for our implementations.

| Methods / # Sup | 250 | 500 | 1,000 | 2,000 | 4,000 |
|---|---|---|---|---|---|
| Pseudo-Label | 49.98 ± 1.17 | 40.55 ± 1.70 | 30.91 ± 1.73 | 21.96 ± 0.42 | 16.21 ± 0.11 |
| Π-Model | 53.02 ± 2.05 | 41.82 ± 1.52 | 31.53 ± 0.98 | 23.07 ± 0.66 | 17.41 ± 0.37 |
| Mean Teacher | 47.32 ± 4.71 | 42.01 ± 5.86 | 17.32 ± 4.00 | 12.17 ± 0.22 | 10.36 ± 0.25 |
| VAT | 36.03 ± 2.82 | 26.11 ± 1.52 | 18.68 ± 0.40 | 14.40 ± 0.15 | 11.05 ± 0.31 |
| MixMatch | 11.08 ± 0.87 | 9.65 ± 0.94 | 7.75 ± 0.32 | 7.03 ± 0.15 | 6.24 ± 0.06 |
| UDA (RandAugment) | **8.76 ± 0.90** | **6.68 ± 0.24** | **5.87 ± 0.13** | **5.51 ± 0.21** | **5.29 ± 0.25** |

Table 7: Error rate (%) for CIFAR-10.

**SVHN**  In Table 8, we similarly show results for compared methods of Figure 4b and results of methods mentioned above. Fully supervised learning using 73,257 examples achieves an error rate of 2.84 and 2.28 with or without RandAugment. The performance of the baseline models are reported by MixMatch (Berthelot et al., 2019). Our reimplementation of MixMatch also resulted in comparable but higher error rates than the reported ones.

| Methods / # Sup | 250 | 500 | 1,000 | 2,000 | 4,000 |
|---|---|---|---|---|---|
| Pseudo-Label | 21.16 ± 0.88 | 14.35 ± 0.37 | 10.19 ± 0.41 | 7.54 ± 0.27 | 5.71 ± 0.07 |
| Π-Model | 17.65 ± 0.27 | 11.44 ± 0.39 | 8.60 ± 0.18 | 6.94 ± 0.27 | 5.57 ± 0.14 |
| Mean Teacher | 6.45 ± 2.43 | 3.82 ± 0.17 | 3.75 ± 0.10 | 3.51 ± 0.09 | 3.39 ± 0.11 |
| VAT | 8.41 ± 1.01 | 7.44 ± 0.79 | 5.98 ± 0.21 | 4.85 ± 0.23 | 4.20 ± 0.15 |
| MixMatch | 3.78 ± 0.26 | 3.64 ± 0.46 | 3.27 ± 0.31 | 3.04 ± 0.13 | 2.89 ± 0.06 |
| UDA (RandAugment) | **2.76 ± 0.17** | **2.70 ± 0.09** | **2.55 ± 0.09** | **2.57 ± 0.09** | **2.47 ± 0.15** |

Table 8: Error rate (%) for SVHN.

## B  ADDITIONAL TRAINING TECHNIQUES

While UDA generally works well, there are several practical issues in consistency training that may dampen the performance gain if not carefully dealt with. This section presents additional techniques targeting at some commonly encountered problems.

**Sharpening Predictions.** Entropy minimization (Grandvalet & Bengio, 2005) has been shown to be effective in semi-supervised learning methods such as VAT (Miyato et al., 2018). To apply entropy minimization in our method, we simply add a loss term to the objective to regularize the predicted distributions on unlabeled examples to have a low entropy. Alternatively, when the number of labeled examples are extremely small, we find it helpful to mask out examples that the current model is not confident about and use a low Softmax temperature when computing the target distribution on unlabeled examples. Specifically, in each minibatch, the consistency loss term is computed only on examples whose highest probability among classification categories is greater than a threshold.

**Domain-relevance Data Filtering.** Ideally, we would like to make use of out-of-domain unlabeled data since it is usually much easier to collect, but the class distributions of out-of-domain data are mismatched with those of in-domain data, which can result in performance loss if directly used (Oliver et al., 2018). To obtain data relevant to the domain for the task at hand, we adopt a common technique for detecting out-of-domain data. We use our baseline model trained on the in-domain data to infer the labels of data in a large out-of-domain dataset and pick out examples that the model is most confident about. Specifically, for each category, we sort all examples based on the classified probabilities of being in that category and select the examples with the highest probabilities.

## C  EXTENDED AUGMENTATION STRATEGIES FOR DIFFERENT TASKS

**Discussion on Trade-off Between Diversity and Validity for Data Augmentation.**  Despite that state-of-the-art data augmentation methods can generate diverse and valid augmented examples as discussed in section 2.2, there is a trade-off between diversity and validity since diversity is

achieved by changing a part of the original example, naturally leading to the risk of altering the ground-truth label. We find it beneficial to tune the trade-off between diversity and validity for data augmentation methods. For text classification, we tune the temperature of random sampling. On the one hand, when we use a temperature of $0$, decoding by random sampling degenerates into greedy decoding and generates perfectly valid but identical paraphrases. On the other hand, when we use a temperature of $1$, random sampling generates very diverse but barely readable paraphrases. We find that setting the Softmax temperature to $0.7, 0.8$ or $0.9$ leads to the best performances.

**RandAugment Details.** In our implementation of RandAugment, each sub-policy is composed of two operations, where each operation is represented by the transformation name, probability, and magnitude that is specific to that operation. For example, a sub-policy can be [(Sharpness, 0.6, 2), (Posterize, 0.3, 9)].

For each operation, we randomly sample a transformation from $15$ possible transformations, a magnitude in $[1, 10]$ and fix the probability to $0.5$. Specifically, we sample from the following 15 transformations: Invert, Cutout, Sharpness, AutoContrast, Posterize, ShearX, TranslateX, TranslateY, ShearY, Rotate, Equalize, Contrast, Color, Solarize, Brightness. We find this setting to work well in our first try and did not tune the magnitude range and the probability. Tuning these hyperparameters might result in further gains in accuracy.

**TF-IDF based word replacing Details.** We describe the TF-IDF based word replacing data augmentation method in this section. Ideally, we would like the augmentation method to generate both diverse and valid examples. Hence, the augmentation is designed to retain keywords and replace uninformative words with other uninformative words. We use BERT's word tokenizer since BERT first tokenizes sentences into a sequence of words and then tokenize words into subwords although the model uses subwords as input.

Specifically, Suppose $\text{IDF}(w)$ is the IDF score for word $w$ computed on the whole corpus, and $\text{TF}(w)$ is the TF score for word $w$ in a sentence. We compute the TF-IDF score as $\text{TFIDF}(w) = \text{TF}(w)\text{IDF}(w)$. Suppose the maximum TF-IDF score in a sentence $x$ is $C = \max_i \text{TFIDF}(x_i)$. To make the probability of having a word replaced to negatively correlate with its TF-IDF score, we set the probability to $\min(p(C - \text{TFIDF}(x_i))/Z, 1)$, where $p$ is a hyperparameter that controls the magnitude of the augmentation and $Z = \sum_i (C - \text{TFIDF}(x_i))/|x|$ is the average score. $p$ is set to $0.7$ for experiments on DBPedia.

When a word is replaced, we sample another word from the whole vocabulary for the replacement. Intuitively, the sampled words should not be keywords to prevent changing the ground-truth labels of the sentence. To measure if a word is keyword, we compute a score of each word on the whole corpus. Specifically, we compute the score as $S(w) = \text{freq}(w)\text{IDF}(w)$ where $\text{freq}(w)$ is the frequency of word $w$ on the whole corpus. We set the probability of sampling word $w$ as $(\max_{w'} S(w') - S(w))/Z'$ where $Z' = \sum_w \max_{w'} S(w') - S(w)$ is a normalization term.

## D    EXTENDED RELATED WORK

**Semi-supervised Learning.** Due to the long history of semi-supervised learning (SSL), we refer readers to (Chapelle et al., 2009) for a general review. More recently, many efforts have been made to renovate classic ideas into deep neural instantiations. For example, graph-based label propagation (Zhu et al., 2003) has been extended to neural methods via graph embeddings (Weston et al., 2012; Yang et al., 2016) and later graph convolutions (Kipf & Welling, 2016). Similarly, with the variational auto-encoding framework and reinforce algorithm, classic graphical models based SSL methods with target variable being latent can also take advantage of deep architectures (Kingma et al., 2014; Maaløe et al., 2016; Yang et al., 2017). Besides the direct extensions, it was found that training neural classifiers to classify out-of-domain examples into an additional class (Salimans et al., 2016) works very well in practice. Later, Dai et al. (2017) shows that this can be seen as an instantiation of low-density separation.

Apart from enforcing consistency on the noised input examples and the hidden representations, another line of research enforces consistency under different model parameters, which is complementary to our method. For example, Mean Teacher (Tarvainen & Valpola, 2017) maintains a teacher model with parameters being the ensemble of a student model's parameters and enforces the consistency between the predictions of the two models. Recently, Athiwaratkun et al. (2018) propose

fast-SWA that improves Mean Teacher by encouraging the model to explore a diverse set of plausible parameters. In addition to parameter-level consistency, SNTG (Luo et al., 2018) also enforces input-level consistency by constructing a similarity graph between unlabeled examples.

**Data Augmentation.** Also related to our work is the field of data augmentation research. Besides the conventional approaches and two data augmentation methods mentioned in Section 2.1, a recent approach MixUp (Zhang et al., 2017) goes beyond data augmentation from a single data point and performs interpolation of data pairs to achieve augmentation. Recently, Hernández-García & König (2018) have shown that data augmentation can be regarded as a kind of explicit regularization methods similar to Dropout.

**Diverse Back Translation.** Diverse paraphrases generated by back-translation has been a key component in the significant performance improvements in our text classification experiments. We use random sampling instead of beam search for decoding similar to the work by Edunov et al. (2018). There are also recent works on generating diverse translations (He et al., 2018; Shen et al., 2019; Kool et al., 2019) that might lead to further improvements when used as data augmentations.

**Unsupervised Representation Learning.** Apart from semi-supervised learning, unsupervised representation learning offers another way to utilize unsupervised data. Collobert & Weston (2008) demonstrated that word embeddings learned by language modeling can improve the performance significantly on semantic role labeling. Later, the pre-training of word embeddings was simplified and substantially scaled in Word2Vec (Mikolov et al., 2013) and Glove (Pennington et al., 2014). More recently, Dai & Le (2015); Peters et al. (2018); Radford et al. (2018); Howard & Ruder (2018); Devlin et al. (2018) have shown that pre-training using language modeling and denoising auto-encoding leads to significant improvements on many tasks in the language domain. There is also a growing interest in self-supervised learning for vision (Zhai et al., 2019b; Hénaff et al., 2019; Trinh et al., 2019).

**Consistency Training in Other Domains.** Similar ideas of consistency training has also been applied in other domains. For example, recently, enforcing adversarial consistency on unsupervised data has also been shown to be helpful in adversarial robustness (Stanforth et al., 2019; Zhai et al., 2019a; Carmon et al., 2019). Enforcing consistency w.r.t data augmentation has also been shown to work well for representation learning (Hu et al., 2017; Ye et al., 2019). Invariant representation learning (Liang et al., 2018; Salazar et al., 2018) applies the consistency loss not only to the predicted distributions but also to representations and has been shown significant improvements on speech recognition.

# E    EXPERIMENT DETAILS

In this section, we provide experiment details for the performed experiments.

## E.1    TEXT CLASSIFICATIONS

**Datasets.** In our semi-supervised setting, we randomly sampled labeled examples from the full supervised set[5] and use the same number of examples for each category. For unlabeled data, we use the whole training set for DBPedia, the concatenation of the training set and the unlabeled set for IMDb and external data for Yelp-2, Yelp-5, Amazon-2 and Amazon-5 (McAuley et al., 2015)[6]. Note that for Yelp and Amazon based datasets, the label distribution of the unlabeled set might not match with that of labeled datasets since there are different number of examples in different categories. Nevertheless, we find it works well to use all the unlabeled data.

**Preprocessing.** We find the sequence length to be an important factor in achieving good performance. For all text classification datasets, we truncate the input to 512 subwords since BERT is pretrained with a maximum sequence length of 512. Further, when the length of an example is greater than 512, we keep the last 512 subwords instead of the first 512 subwords as keeping the latter part of the sentence lead to better performances on IMDb.

---

[5]http://bit.ly/2kRWoof, https://ai.stanford.edu/~amaas/data/sentiment/
[6]https://www.kaggle.com/yelp-dataset/yelp-dataset, http://jmcauley.ucsd.edu/data/amazon/

**Fine-tuning BERT on in-domain unsupervised data.** We fine-tune the BERT model on in-domain unsupervised data using the code released by BERT. We try learning rate of 2e-5, 5e-5 and 1e-4, batch size of 32, 64 and 128 and number of training steps of 30k, 100k and 300k. We pick the fine-tuned models by the BERT loss on a held-out set instead of the performance on a downstream task.

**Random initialized Transformer.** For the experiments with randomly initialized Transformer, we adopt hyperparameters for BERT base except that we only use 6 hidden layers and 8 attention heads. We also increase the dropout rate on the attention and the hidden states to 0.2, When we train UDA with randomly initialized architectures, we train UDA for 500k or 1M steps on Amazon-5 and Yelp-5 where we have abundant unlabeled data.

**BERT hyperparameters.** Following the common BERT fine-tuning procedure, we keep a dropout rate of 0.1, and try learning rate of 1e-5, 2e-5 and 5e-5 and batch size of 32 and 128. We also tune the number of steps ranging from 30 to 100k for various data sizes.

**UDA hyperparameters.** We set the weight on the unsupervised objective $\lambda$ to 1 in all of our experiments. We use a batch size of 32 for the supervised objective since 32 is the smallest batch size on v3-32 Cloud TPU Pod. We use a batch size of 224 for the unsupervised objective when the Transformer is initialized with BERT so that the model can be trained on more unlabeled data. We find that generating one augmented example for each unlabeled example is enough for $\text{BERT}_{\text{FINETUNE}}$.

All experiments in this part are performed on a v3-32 Cloud TPU Pod.

### E.2 Semi-supervised learning benchmarks CIFAR-10 and SVHN

**Hyperparameters for Wide-ResNet-28-2.** For hyperparameter tuning, for simplicity, we performed a random sampling search over hyperparameters and choose the best one based on validation sets (20% of the training sets with different sizes). We use the averaged results of multiple experiments to reduce the performance variance measured on the small validation sets. Specifically, we tried the following ranges:

- training steps: 50k, **100k**;
- learning rate: **0.03**, 0.05, 0.1;
- TSA schedules: log-schedule, **linear-schedule**, exp-schedule, not using TSA;
- entropy minimization loss weight: 0, **0.1**, 0.3;
- consistency loss weight: **1**, 3, 6;
- unlabeled data batch size: **960**, 1280;
- weight decay rate: **5e-4**, 7e-4, 1e-3;
- softmax temperature: **1**, 0.9;
- confidence threshold: **0**, 0.8.

where the values in bold black text are our default hyperparameters. We found that given a reasonably large labeled set, our method is robust to hyper-parameters. Therefore, we use the same hyper-parameters in these cases. Specifically, we use the above default hyperparameters for CIFAR-10 with 4,000, 2,000, 1,000 and 500 examples. For SVHN with 4,000, 2,000, 1,000, 500, 250 examples, we additionally set the learning rate to 0.05 and unlabeled batch size to 1280. For the case of 250 examples on CIFAR-10, we use a different set of hyper-parameters: training steps: 50k; TSA schedule: log-schedule; consistency loss coefficient: 6; weight decay: 7e-4; unlabeled data batch size: 1280; softmax temperature: 0.9; consistency threshold: 0.8.

Other hyperparameters not mentioned above are the same to the the original paper of Wide-ResNet (Zagoruyko & Komodakis, 2016). In order to reduce training time, we generate augmented examples before training and dump them to disk. For CIFAR-10, we generate 100 augmented examples for each unlabeled example. Note that generating augmented examples in an online fashion is always better or as good as using dumped augmented examples since the model can see different augmented examples in different epochs, leading to more diverse samples. We report the average performance and the standard deviation for 10 runs.

**Hyperparameters for Shake-Shake and PyramidNet.** For the experiments with Shake-Shake, we train UDA for 300k steps and use a batch size of 128 for the supervised objective and use a batch

size of 512 for the unsuperivsed objective. For the experiments with PyramidNet+ShakeDrop, we train UDA for 700k steps and use a batch size of 64 for the supervised objective and a batch size of 128 for the unsupervised objective. For both models, we use a learning rate of 0.03 and use a cosine learning decay with one annealing cycle following AutoAugment.

All experiments in this part are performed on a v3-32 Cloud TPU v3 Pod.

### E.3 IMAGENET

**10% Labeled Set Setting.** Unless otherwise stated, we follow the standard hyperparameters used in an open-source implementation of ResNet.[7] For the 10% labeled set setting, we use a batch size of 512 for the supervised objective and a batch size of 15,360 for the unsupervised objective. We use a base learning rate of 0.3 that is decayed by 10 for four times and set the weight on the unsupervised objective $\lambda$ to 20. We mask out unlabeled examples whose highest probabilities across categories are less than 0.5 and set the Softmax temperature to 0.4. The model is trained for 40k steps. Experiments in this part are performed on a v3-64 Cloud TPU v3 Pod.

**Full Labeled Set Setting.** For experiments on the full ImageNet, we use a batch size of 8,192 for the supervised objective and a batch size of 16,384 for the unsupervised objective. The weight on the unsupervised objective $\lambda$ is set to 1. We use entropy minimization to sharpen the prediction. We use a base learning rate of 1.6 and decay it by 10 for four times. Experiments in this part are performed on a v3-128 Cloud TPU v3 Pod.

---

[7]https://github.com/tensorflow/tpu/tree/master/models/official/resnet

