# OpenReview forum: "Unsupervised Data Augmentation for Consistency Training"
_ICLR.cc/2020/Conference — Reject_

### Official Review · AnonReviewer1 · 2019-10-20
**Official Blind Review #1**

**Rating:** 8

**Review:**

The paper "Unsupervised Data Augmentation for Consistency Training" marries two recent ideas of
1. "Data Augmentation" (DA) from supervised learning: The authors explore various methods for "DA" mostly inspired by much recent work such as Random image transformations, Backtranslation, and TF-IDF based word replacement.
2. "Consistency Training" (CT) from semi-supervised learning: CT tries to minimize the divergence between the output distributions of the classifiers that are produced by adding noise to the input.

The key insight in this paper is that, data augmentation methods that work well during supervised training should also work equally well as the noise distribution for consistency training on unlabeled data. The authors support this claim empirically through the experiments in table 1 and 2.

The paper is well written and the authors present extensive comparative and ablation tests to demonstrate that their proposed method works well with both low and high amounts of labeled data.  This paper should be accepted into the conference.

**Experience Assessment:**

I have read many papers in this area.

**Review Assessment: Checking Correctness Of Derivations And Theory:**

N/A

**Review Assessment: Checking Correctness Of Experiments:**

I assessed the sensibility of the experiments.

**Review Assessment: Thoroughness In Paper Reading:**

I read the paper thoroughly.

---

> ### Author Response · Authors · 2019-11-11
> **Thank you for your valuable feedback!**

---

### Official Review · AnonReviewer2 · 2019-10-23
**Official Blind Review #2**

**Rating:** 3

**Review:**

In this paper, the authors present a new perspective on how to effectively noise unlabeled examples and argue that the quality of noising plays a crucial role in semi-supervised learning. By substituting simple noising operations with advanced data augmentation methods, their method brings substantial improvements across six language and three vision tasks under the same consistency training framework. I think the topic itself is interesting and I have the following concerns.
(1) The first is about the contribution of this paper. In this paper, all the results, including the augmented methods are all well established approaches. The authors have just employed them in solving a new problem, without support about why they work. Thus, the results are only strategies, without theoretical guarantee or insights. It is difficult to convince the reviewers.
(2) Although the authors have achieved seemingly promising results, I think it can not convince me since the authors have not answered the questions about why and when. I think this paper likes a technical report, not a research paper.
(3) I have also noticed the discussions among the authors and other readers. It seems that the large improvement depends on the parameters heavily. So, why not to share the parameters directly?

**Experience Assessment:**

I have read many papers in this area.

**Review Assessment: Checking Correctness Of Derivations And Theory:**

I assessed the sensibility of the derivations and theory.

**Review Assessment: Checking Correctness Of Experiments:**

I assessed the sensibility of the experiments.

**Review Assessment: Thoroughness In Paper Reading:**

I made a quick assessment of this paper.

---

> ### Author Response · Authors · 2019-11-11
> **Thank you for your valuable feedback!**
>
> [Contribution]
> It is not known before that the quality of data augmentation can lead to such a large and consistent gain on the semi-supervised learning performance. Our work is largely empirical and we have conducted extensive experiments to support our results.
>
> [Hyper-parameters]
> We only use a different hyper-parameter for the case of 250 examples on CIFAR-10. The hyper-parameters for other data sizes are the same. We found that given a reasonably large labeled set, our method is robust to hyper-parameters. Therefore, we use the same hyper-parameters in other cases.
>
> We have already uploaded the hyper-parameters to github at the request of the comment but did not provide a link here to preserve anonymity.
>
> Our default hyper-parameters are as follows:
> * training steps: 100k;
> * learning rate: 0.03;
> * TSA schedule: linear-schedule;
> * entropy minimization loss weight: 0.1;
> * consistency loss coefficient: 1;
> * weight decay: 5e-4;
> * unlabeled data batch size: 960;
> * softmax temperature: 1 (not used);
> * consistency threshold: 0 (not used).
>
> Additional hyper-parameters for SVHN with 4,000, 2,000, 1,000, 500, 250 examples: learning rate: 0.05; unlabeled data batch size: 1280.
>
> Additional hyper-parameters for CIFAR-10 with 4,000, 2,000, 1,000, 500 examples: no additional hyper-parameters.
>
> Additional hyper-parameters for CIFAR-10 with 250 examples:
> * training steps: 50k;
> * TSA schedule: log-schedule;
> * consistency loss coefficient: 6;
> * weight decay: 7e-4;
> * unlabeled data batch size: 1280;
> * softmax temperature: 0.9;
> * consistency threshold: 0.8.
>
> We have included these detailed hyper-parameters in our paper.

---

> > ### Author Response · Authors · 2019-11-13
> > **Please consider increasing your score if our response clarifies.**
> >
> > As we have addressed your concerns about hyper-parameters, we hope the reviewer could reconsider the score. Thank you again for your suggestions to help us improve the paper!

---

### Official Review · AnonReviewer3 · 2019-10-23
**Official Blind Review #3**

**Rating:** 3

**Review:**

The paper proposes to substitute simple noising operations with many data augmentation methods in consistency-based semi-supervised learning. The main idea is the same as previous work: constrain the model predictions of unlabeled examples to be invariant to different noise. The proposed UDA is evaluated on a wide range of language and vision tasks.

Overall, the paper is well-written and clear. The most impressive point of this paper is its strong empirical results. However, it looks not surprising to me that more data augmentations found in supervised learning are also effective in semi-supervised learning. The paper fails to provide any theoretical insights but a thorough empirical evaluation.

One of my concerns is that the hyperparameters on vision tasks follow those of AutoAugment, which is carefully tuned on supervised tasks. Apparently, their hyperparameters are based on the whole labeled training dataset. In this case, the adopted hyperparameters include sort of information of the whole labeled dataset. Is it fair?

Another concern is how to control the strength of augmentations. For example, for digit images like SVHN, a "6" rotates by 180 degree is "9", whose prediction should change correspondingly. In this case, the assumption of invariance does not hold when the augmentation is too strong.

I'm willing to increase my score if the authors address my concerns.

**Experience Assessment:**

I have published one or two papers in this area.

**Review Assessment: Checking Correctness Of Derivations And Theory:**

I carefully checked the derivations and theory.

**Review Assessment: Checking Correctness Of Experiments:**

I carefully checked the experiments.

**Review Assessment: Thoroughness In Paper Reading:**

I read the paper thoroughly.

---

> ### Author Response · Authors · 2019-11-11
> **Thank you for your valuable feedback!**
>
> [Contribution]
> It is not known before that the quality of data augmentation can lead to such a large and consistent gain on the semi-supervised learning performance. Our work is largely empirical and we have conducted extensive experiments to support our results.
>
> [Strengths of augmentation]
> For each transformation in RandAugment, there is a single scalar from 1 to 10 to control the strength of the augmentation (higher value means more changes to the content of the input image). In theory, it is possible to adjust the distribution or range of the augmentation strength on each dataset based on dev performance. However, in this work, we find it works well to simply uniformly sample from 1 to 10 without any tuning.
>
> [Hyper-parameters]
> For simplicity, we performed a random sampling search over hyper-parameters and choose the best one based on validation sets (20% of the training sets with different sizes). Specifically, we tried the following ranges:
> * training steps: 50k, 100k;
> * learning rate: 0.03, 0.05, 0.1;
> * TSA schedules: log-schedule, linear-schedule, exp-schedule, not using TSA;
> * entropy minimization loss weight: 0, 0.1, 0.3;
> * consistency loss weight: 1, 3, 6;
> * weight decay rate: 5e-4, 7e-4, 1e-3;
> * unlabeled data batch size: 960, 1280;
> * softmax temperature: 1, 0.9;
> * confidence threshold: 0, 0.8.
>
> The sentence in the Appendix ''Other hyper-parameters follow those of the released AutoAugment code'' actually refers to the fact that we employ the same model architectures, including Wide-ResNet, Shake-Shake and ShakeDrop, as in the AutoAugment paper. AutoAugment use the same hyper-parameters as reported in the papers introducing the models except for the weight decay and the learning rate schedule.
>
> Since we use RandAugment instead of AutoAugment to produce perturbation, we DO NOT rely on any augmentation strategy searched by AutoAugment at all.

---

> > ### Author Response · Authors · 2019-11-13
> > **Please consider increasing your score if our response clarifies.**
> >
> > As we have addressed your concerns about hyper-parameters and augmentation strengths, we hope the reviewer could reconsider the score, as promised in the original review. Thank you again for your suggestions to help us improve the paper!

---

### Public Comment · ~Xiao_Wang6 · 2019-10-21
**Doubted about the results**

I think KL divergence is not a new idea in your paper for the semi-supervised area, which has been proposed in VAT. I don't know why your results work so well.
Therefore, I simply run your code on cifar with the command you suggested. The results listed in the paper can't be achieved for 250 label, 500 label, 1000label. I think you may fine tuned the hyper parameters to get the results.
However, in semi-supervised setting, for different number of labels, you must used the same hyper-parameters to evaluate the results.
Also, please release the command to run your code for different labels and show your training loss/accuracy vs epoch figures for 250labels.

---

> ### Author Response · Authors · 2019-10-21
> **Source of Effectiveness.**
>
> Hi, the effectiveness of our method is due to employing advanced data augmentation instead of using KL divergence. Specifically, we found that state-of-the-art data augmentations found in supervised learning can also serve as a superior source of noise under the consistency enforcing semi-supervised framework. The importance of advanced data augmentation is demonstrated clearly in Section 3.1, i.e., our study of correlation between supervised and semi-supervised performances. When we use cropping and flipping to augment the data, we only achieve an error rate of 16.17 with 4,000 labeled examples. In contrast, using RandAugment leads to an error rate of 4.23.
>
> We followed prior works on semi-supervised learning (e.g., SNTG https://arxiv.org/pdf/1711.00258.pdf) and tuned hyperparameters for different data sizes. We do strictly limit the number of labeled examples while tuning the hyperparameters. For example, to determine the best hyperparameter for the setting with 250 labeled examples, we used 200 examples as the training set and 50 examples as the validation set and found out the best hyperparameters. Then we used the found hyperparameter to train a model with 250 labeled examples. We will release the corresponding hyperparameters.

---

> > ### Public Comment · ~Xiao_Wang6 · 2019-11-01
> > **How can we believe your results**
> >
> > You said you use 50 examples to pick best hyper-parameter. This is not included in your code and your paper. Also, i never think 50 examples can really reflect the real distribution of the 50,000 examples. That is completely random if you use different random seed. I should say, if you tune hyper parameter based on that, what you get is completely random. Furthermore, you said you tune that based on 50 examples. Then please give me the code, let me run with different random seed to pick hyper-parameter and I am sure they are not the same and lead to completely different results. I am very confident you can not even achieve the mixmatch's performance.
> > In your mentioned paper, they are based 4,000 images. You use only 50 image to pick up best hyper-parameter. From my experience, that's impossible. If that can work, please release your code to let everyone
> > to have a test.

---

> > > ### Author Response · Authors · 2019-11-04
> > > **You can tune hyperparameters with a small dev set by running experiments multiple times to reduce variance.**
> > >
> > > Thank you for your comments. First, we actually do not need to tune hyperparameters for a specific labeled set size except for 250 examples case on CIFAR-10. The same or similar hyperparameters work well for the cases with 500, 1000, 2000, 4000 examples on CIFAR-10 and work well across all data sizes for SVHN. As for the case with 250 examples on CIFAR-10, using the same hyperparameter leads to an error rate of $16.8 \pm 4.19$. We will include the performance without hyperparameter tuning into the results section of our paper and include the tuned hyperparameters into the Appendix.
> > >
> > > We do not yet know why using 250 labeled examples requires a very different hyperparameter. My guess is that it is a stability issue and improving the implementation's stability could help it. For example, it might help to employ Exponential Moving Average of model parameters, using a different learning rate schedule and so on. I personally do not have time to investigate it at this moment due to other obligations such as preparing thesis. If you indeed care about the case with a very small number of examples, it might be interesting to study how to improve the stability.
> > >
> > > Second, for the 250 examples case, we can achieve better performance by tuning hyperparameters and it is feasible to tune hyperparameters with a small number of dev examples, e.g., with 50 examples. The key is to run experiments multiple times and use the average accuracy instead of a single accuracy to determine which hyperparameter is better.
> > >
> > > Empirically, with 50 images in the dev set, it works well to run 10-20 experiments for each hyperparameter. Performance of semi-supervised learning algorithms is usually reported on 10 runs [1, 2, 3], so running multiple times should be expected though it incurs a significant computation cost. We will upload the code for splitting data so that you can try it yourself. We will also upload logs of tuning hyperparameters for the 250 labels' case.
> > >
> > > Theoretically, there are two difficulties to overcome when using a small dev set: (1) the hyperparameter that work well for a small dev set can overfit the small set and does not work well for a larger dev / test set. We empirically verified that this situation does not hold. We compared hyperparameter selection using a fixed small dev set and using cross validation and found that they lead to similar selections of hyperparameters. In fact, it has been found that models which performs well on a fixed test set generalize to other test sets [4, 5]. (2) It can be hard to estimate the performance on a small dev set due to a large variance. This difficulty is real and we run experiments multiple times and take the average to reduce the variance.
> > >
> > > Central Limit Theorem (CLT) can explain why it helps to run experiments multiple times. This is a self-contained description of CLT: Without loss of generality, suppose that for a set of hyperparameter $h$, the accuracy measured on $k$ dev samples is subject to a Gaussian distribution
> > > $$\mathcal{N}(\mu_{h}, \sigma_{h, k}^2)$$
> > > where the mean $\mu_{h}$ is dependent on the hyperparameter $h$ and the standard deviation $\sigma_{h, k}$ depends on both $h$ and $k$. A better hyperparameter would lead to a higher mean $\mu_{h}$. A smaller dev set size would lead to a larger std $\sigma_{h, k}$.
> > >
> > > To determine whether $h_1$ is better than $h_2$, we run $n$ experiments for both $h_1$ and $h_2$. Suppose we get accuracies $X_1, X_2, \cdots, X_n$ for hyperparameter $h_1$ and $Y_1, Y_2, \cdots, Y_n$ for hyperparameter $h_2$. We compute their sample mean by  $$\bar{X}=\frac{1}{n}\sum_{i=1}^n X_i$$ $$\bar{Y}=\frac{1}{n}\sum_{i=1}^n Y_i$$
> > >
> > > Basically, $\bar{X}$ and $\bar{Y}$ are our estimates for $\mu_{h_1}$ and $\mu_{h_2}$ and we decide that $h_1$ is better than $h_2$ if $\bar{X}$ is higher than $\bar{Y}$.
> > >
> > > By CLT, we have that $\bar{X}$ and $\bar{Y}$ converge in distribution to $$\mathcal{N}(\mu_{h_1}, \sigma_{h_1, k}^2 / n)$$ $$\mathcal{N}(\mu_{h_2}, \sigma_{h_2, k}^2 / n)$$
> > >
> > > So when $\sigma_{h_1, k}$ and $\sigma_{h_2, k}$ are large due to a small dev size, if you just run one experiment for a set of hyperparameter (setting $n$ to $1$), then the variance of $\bar{X}$ and $\bar{Y}$ might dominate the difference in $\mu_{h_1}$ and $\mu_{h_2}$. But if you sample multiple times by setting $n$ to $10$ or a larger number, the variance of $\bar{X}$ and $\bar{Y}$ can be reduced by $n$. So $\bar{X}$ and $\bar{Y}$ are reliable estimates for $\mu_{h_1}$ and $\mu_{h_2}$.
> > >
> > > Third, you mentioned that “In your mentioned paper, they are based 4,000 images. You use only 50 images to pick up the best hyper-parameter”. This is not correct. We said that we use 50 examples for the case with 250 examples. For the case with 4,000 examples, we use 800 of them as the dev set and use the rest 3,200 images for training.
> > >
> > > Let us know if you have further questions or concerns.
> > >
> > > References are listed in the following post due to space limits.

---

> > > > ### Author Response · Authors · 2019-11-04
> > > > **References for the previous post**
> > > >
> > > > References:
> > > >
> > > > [1] Miyato, T., Maeda, S. I., Koyama, M., Ishii, S. (2018). Virtual adversarial training: a regularization method for supervised and semi-supervised learning. IEEE transactions on pattern analysis and machine intelligence, 41(8), 1979-1993.
> > > >
> > > > [2] Tarvainen, A., Valpola, H. (2017). Mean teachers are better role models: Weight-averaged consistency targets improve semi-supervised deep learning results. In Advances in neural information processing systems (pp. 1195-1204).
> > > >
> > > > [3] Laine, S., Aila, T. (2016). Temporal ensembling for semi-supervised learning. arXiv preprint arXiv:1610.02242.
> > > >
> > > > [4] Recht, B., Roelofs, R., Schmidt, L., Shankar, V. (2018). Do CIFAR-10 classifiers generalize to CIFAR-10?. arXiv preprint arXiv:1806.00451.
> > > >
> > > > [5] Recht, B., Roelofs, R., Schmidt, L., Shankar, V. (2019). Do ImageNet Classifiers Generalize to ImageNet?. arXiv preprint arXiv:1902.10811.

---

> > > > > ### Public Comment · ~Xiao_Wang6 · 2020-01-23
> > > > > **Current much more reasonable results**
> > > > >
> > > > > The author just updated their running commands. A script is uploaded to reproduce the CIFAR-10 performance on GPU with a unified hyperparameter for all data sizes. Link: https://github.com/google-research/uda/blob/master/image/scripts/run_cifar10_gpu.sh. I think this used same hyper-parameter to get reasonable results. I should say, now it's a good improvement for the semi-supervised area.

---

### Public Comment · ~Varun_Nair1 · 2019-10-24
**Question about Intuition behind Training Signal Annealing (TSA)**

Hello,

In your paper you provide intuition for when to use different schedules of TSA, specifically the logarithmic, linear, and exponential schedules:

"Intuitively, when the model is prone to overfit, e.g., when the problem is relatively easy or the number of labeled examples is very limited, the exp-schedule is most suitable as the supervised signal is mostly released at the end of training. In contrast, when the model is less likely to overfit (e.g., when we have abundant labeled examples or when the model employs effective regularization), the log-schedule can serve well."

The above would suggest that in the lowest data regimes in your paper (For example: CIFAR10 on 250 labels or SVHN on 250 labels), the exp-schedule would work best and in the higher data regimes the linear and log-schedule would work better.

However, in your code implementation, you have listed that the training signal annealing schedule that achieved SOTA performance on CIFAR10 with 250 labels is the log-schedule.

This would directly contradict what is stated as intuition in the paper, as in the lowest data regime (250 labels) you used the log-schedule, which the paper states as only used "when we have abundant labeled examples". Is this a mistake in the paper or the implementation?

Thank you for clarifying!

---

> ### Author Response · Authors · 2019-10-25
> **Intuition behind TSA**
>
> Hi, good question! Firstly, when entropy minimization is not employed, TSA with exp-schedule indeed works better for small labeled data. As shown in the ablation study for Yelp-5 in Table 6, the model achieves an error rate of 41.35, 45.41, 49.06 with exp-schedule, linear-schedule and log-schedule respectively. Secondly, when entropy minimization is employed, the confidence on labeled data is high even with an exp-schedule, since the model is regularized to make sharp predictions. In this case, TSA with exp-schedule is not able to limit the confidence on labeled data. As for CIFAR-10 with 250 examples, we employed entropy minimization to make the predictions sharper. Hence, we do not employ exp-schedule. We leave it as a future work to study how to better couple TSA with entropy minimization.

---

### Public Comment · ~Zhang_Yuanyuan1 · 2019-11-07
**Doubts about UDA called the methodology of semi-supervised learning**

Hello，
     I have one question about the UDA on Bert as semi-supervised learning method.In the paper ,you said that "on binary sentiment analysis tasks, with only 20 supervised examples, UDA outperforms the previous SOTA trained with full supervised data on IMDb and is competitive on Yelp-2 and Amazon-2."，but in your code I realised that you use the pre-trained model of back translation,which trained on all labeled dataset.As defined in wikipedia——Semi-supervised learning is a class of machine learning tasks and techniques that also make use of unlabeled data for training – typically a small amount of labeled data with a large amount of unlabeled data.In my own opinion,during the training of SSL, it can't use external information. So I think the UDA on NLP tasks couldn't be called as the methodology of semi-supervised learning.
    About UDA on image classification,I'm not very familiar about tasks on image classification. But I read the paper of Autoaugment  previously. Well，I noticed that in the paper you just use a random method to augment data then get  perfect results. As we know that Autoaugment is a very effective method on data augmentation.So is such an impressive Autoaugment  on labeled datasets not as good as a random selection strategy algorithm on a large number of unlabeled datasets?What I mean is that I doubt about the results of UDA on image tasks.

---

> ### Author Response · Authors · 2019-11-11
> **On using Back-translation and RandAugment.**
>
> Thank you for your comments!
>
> [Back translation]
> We acknowledge your point about back-translation using translation data. However, as mentioned in the paper, we would like to point out that the translation task itself is quite distinctive from a text classification task and does not make use of any text classification label. In addition, back-translation is a general data augmentation method that can be applied to many tasks with the same model checkpoints.
>
> [RandAugment]
> AutoAugment is indeed very effective, but RandAugment [1] achieves as good performance and is simpler and more widely applicable to new tasks. Its effectiveness can be attributed to the richness of the augmentations. In Appendix A.1, we show that the diversity of the transformations in RandAugment is essential for its great performance. In addition, we also found that using a fixed set of 25 randomly sampled sub-policies does not achieve as good results.
>
> References:
>
> [1] The paper is available on arXiv, we do not provide a link here to preserve anonymity.

---

### Author Response · Authors · 2019-11-11
**Hyper-parameters used in the paper.**

We list our hyper-parameters here for reproducibility.

Our default hyper-parameters are as follows:
* training steps: 100k;
* learning rate: 0.03;
* TSA schedule: linear-schedule;
* entropy minimization loss weight: 0.1;
* consistency loss coefficient: 1;
* weight decay: 5e-4;
* unlabeled data batch size: 960;
* softmax temperature: 1 (not used);
* consistency threshold: 0 (not used).

Additional hyper-parameters for SVHN with 4,000, 2,000, 1,000, 500, 250 examples: learning rate: 0.05; unlabeled data batch size: 1280.

Additional hyper-parameters for CIFAR-10 with 4,000, 2,000, 1,000, 500 examples: no additional hyper-parameters.

Additional hyper-parameters for CIFAR-10 with 250 examples:
* training steps: 50k;
* TSA schedule: log-schedule;
* consistency loss coefficient: 6;
* weight decay: 7e-4;
* unlabeled data batch size: 1280;
* softmax temperature: 0.9;
* consistency threshold: 0.8.

We only use a different hyper-parameter for the case of 250 examples on CIFAR-10. The hyper-parameters for other data sizes are the same. We found that given a reasonably large labeled set, our method is robust to hyper-parameters. Therefore, we use the same hyper-parameters in other cases.

We have also included these detailed hyper-parameters in our paper.

---

### Decision · Program_Chairs · 2019-12-19

**Decision:**

Reject

**Comment:**

The paper shows that data augmentation methods work well for consistency training on unlabeled data in semi-supervised learning.

Reviewers and AC think that the reported experimental scores are interesting/strong, but scientific reasoning for convincing why the proposed method is valuable is limited. In particular, the authors are encouraged to justify novelty and hyper-parameters used in the paper. This is because I also think that it is not too surprising that more data augmentations in supervised learning are also effective in semi-supervised learning. It can be valuable if more scientific reasoning/justification is provided.

Hence, I recommend rejection.

---

> ### Author Response · Authors · 2020-02-03
> **Response**
>
> We are disappointed that the area chair decided to reject the paper. We believe that novelty is subjective and our work is novel. The concern about hyperparameters is also pretty strange, but is now resolved. See below: https://openreview.net/forum?id=ByeL1R4FvS&noteId=9ruUTMmdG4
>
> A script to reproduce our performance on GPUs using a unified set of hyperparameters can be found here:
>
> CIFAR-10: https://github.com/google-research/uda/blob/master/image/scripts/run_cifar10_gpu.sh
> SVHN: https://github.com/google-research/uda/blob/master/image/scripts/run_svhn_gpu.sh